# Kinesthetic Experience: Emancipatory Corporeal Scores

Celia Vara Martín 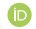

Department of English, McGill University, Montreal, QC H3A 0G5, Canada; celia.vara@mcgill.ca

**Abstract:** This article investigates the corporeal practices by the Catalonian artist Fina Miralles (b.1950) in some of her performances during the 1970s. I specifically focus my analysis on the manner in which the artist verifies the existence of her body under the acute political restrictions on the body during the last years of Francoism (1939–1975). I argue that she does this by a process of sensorial investigations, which include painting, filming and touching natural elements, and moving them and leaving different types of tracks, which lead to generating corporeal scores and body mapping. I elaborate on the way that producing corporeal sensorial knowledge generated from her body mapping and kinaesthetic knowledge is a transgressive and emancipatory feminist intervention. My argument is that kinesthesia generates a process of body-mapping awareness within the body and its movement, which reinforces a sensorial way of knowledge that leads to a reconstitution of the body that function as corporeal agency. Based on feminist theories of embodiment and agency, taking the Carrie Noland concept of kinesthesia (2009) as a central analytical tool, and with a background as a psychologist, I approach this research with embodied methodologies (conversations with the artist, recreation of her actions, etc.) and draw mainly from research-creation methods and kinaesthetic empathy.

**Keywords:** kinaesthesia; kinaesthetic empathy; feminist intervention; sensorial knowledge; embodiment; corporal agency; embodied methodologies; research-creation

La meva obra no és meva; es un art vivencial per a tothom. El conjunct de la meva obra es una llabor que vull que s'excampi, que sigui una motivació per als altres per fer el mateix viatge interior que vaig fer jo.

(My work is not mine; it is an experience-based art. My oeuvre as a while is a seed that I want to spread, which is a motivation for others to do the same inner journey that I did)

Fina Miralles

(personal conversation, 2018) [my translation]

## 1. Introduction

On January 1976, two months after the death of dictator Francisco Franco,[1] Catalonian artist Fina Miralles performed and filmed *Petjades* (Footprints) (Figure 1). This performance consisted of walking in the city streets of Barcelona while she wore shoes that printed the letters of her name on the pavement. The shoes were handmade by Fina Miralles. She put together cork and foam to build up the words "Fina" on one sole and "Miralles" on the other.

In the performed action there was a tray with ink, and she would step in the tray to ink her shoes and start walking, thereby printing her name on the ground. *Petjades* (Figures 1 and 2) was a performance, though its final destination was to be part of a film, which ultimately would include an audio reflection on patrilineal heritage and private property regulations, some of the issues that especially affected women's status in society during the Franco dictatorship.

Although this performance is driven toward the symbolic and the discursive (the imprint of the name, the trace) and proposes to reflect on patriarchal society and private property—highlighting women's situation in an explicit manner through her voice over—the marks

made by Fina Miralles could be understood as governed by a different kind of exploratory movement. When Fina Miralles walks with these shoes, I observe in the Super 8 film how she walks carefully and with a slow movement, feeling the contact of every part of the foot, and then moving the other foot to make a step. She had to walk slowly because the shoes were delicate, and she wanted to have a clean imprint on the ground. However, this fact encourages a bodily movement that could be linked with the sensorial explorations of her previous works (Vara 2021a).

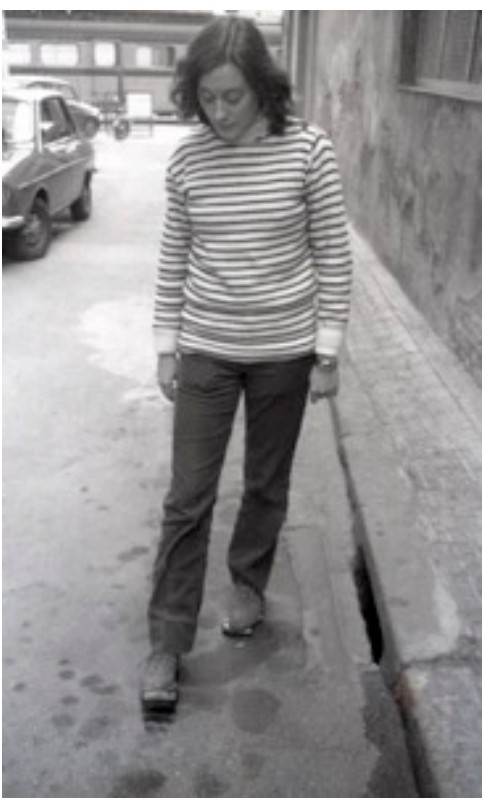

**Figure 1.** Fina Miralles touches the ground carefully to print her name on it. Documentation in black and white photograph of *Petjades* (Footprints) (1976) by Fina Miralles (1976). Photograph by Carles Raurich. Reproduced with permission from the artist and *Museu d'Art de Sabadell* (MAS) archive.

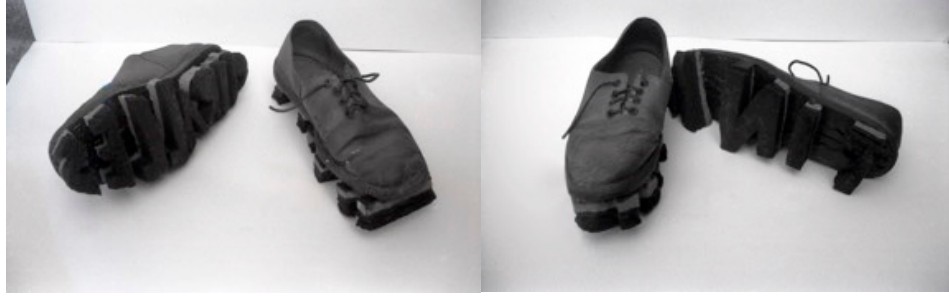

**Figure 2.** Shoes used in *Petjades* (Footprints) with foam attached to the soles with the name of Fina on one sole and Miralles on the other (1976) by Fina Miralles. Reproduced with permission from the artist and *Museu d'Art de Sabadell* (MAS) archive.

What was specific about her work that provided a model for feminist emancipation in a different sphere from the arena of representational democracy? Demonstrations in the streets, mobilizing for women's rights, and feminism's emphasis on consciousness-raising?[2] How was the work on and in the self a mode of the political in its very origins? How was this a different kind of liberatory practice? How were Fina Miralles' corporeal

dynamics a kind of corporeal movement that values the somatic sensation as a site for knowledge production and political intervention? These questions are drawn from the feminist thought on performance actions as liberatory/emancipatory (Jones 2020). I reflect on Fina Miralles for the particular corporeality in her work as a way to glean knowledge within the body and from the work, under Francoism.[3] Feminist theories of liberation have examined concepts of agency and embodiment, and state the human body to be a means for agency, and the world around us as something that can stimulate and also restrain agency (McNay 2000; Meynell 2009; Coole 2005; Krause 2011; Gilligan 2016). Consequently, human agency has a corporeal meaning and can be understood as an agent that transforms its relation with itself and with itself and with the world.

In the following, I argue for the senses as a source for confirming the existence of a body through performance work—in such acute politics—and as a manner to generate knowledge specifically through the sense of kinesthesia. Drawing from the dance and performance scholar Carrie Noland, I define kinesthesia as the sense of our body position and movement in a given space (Noland 2009). She states that "experience—the sensory awareness of one's own movement—can indeed encourage experiment, modification, and, at times, rejection of the routine" (Noland 2009, p. 3). Thus, kinesthesia is a potential somatic tool for emancipation processes because it is a way to access corporeal knowledge of the self and the surroundings. I work closely with the notion of kinesthetic empathy that can engage in and with another's movement or sensorial experience of movement (Reynolds and Reason 2012; Sklar 1994, 2008). That allows experimentation and creation of new bodily routines and, therefore, a form of agency that permits the exploration of subtle corporeal resistance in conditions of body repression/restrictions such as dictatorships.

In this article I elaborate on the somatic strategies in *Petjades*, and some of her earlier works (*Sensitive Land*, *Natura Morta* (Still Life), *Fenòmens Atmosfèrics* (Atmospherical Phenomena), *Translacions* (Movements)[4] and *Relacions*[5] (Relations)), as sensorially instructive for a feminist praxis. I decided to approach and combine these works because this allows me to see kinesthetic and sensorial aspects in *Petjades* that have been missed in that action because of its political and semiotic valence. Her corporeal practices and protofeminist work help her to draw the borders of the body, its silhouette, create tracks and leave footprints/marks that would serve as somatic instructions for her more politically explicit works[6] developed later after Franco's death. They conduct an introspective activity and assist to confirm the existence of a body that occupies a space—a body lost in dictatorship and its aftermaths—and provide corporeal scores for subsequent generations.

In the process of this research I asked myself: how can I engage with her body-work? How can my methodology be coherent with a research question that centres on corporeal dynamics? Could I develop a tactile and corporeal sense of her practice in and through my research? Based on feminist theories of embodiment and agency, and taking Carrie Noland's (2009) concept of kinesthesia as a central analytical tool, I approach this investigation through research-creation methodologies based on kinesthetic empathy. Research-creation is a category within social sciences and humanities that integrates creativity, artistic work, practices and experiences as part of the research (Chapman and Sawchuk 2012). This methodology "incites us to inquire into the very question of how practices produce knowledge" (Manning 2016, p. 26).

My training as a psychologist shaped the approach I used in this research. I used my abilities in interviewing, listening, observation and empathy as modes of research performance art and analysis of body movement and moving images. This research into embodiment was experimental in its methodologies: I made use of my own performative actions (Vara 2022), re-creations of Miralles' performance in the sites where they took place, filming, and extensive archival work in Spain—which included personal conversations with the artist (Miralles 2016–2019), interviews with Catalonian art historians and artists related to the artist, analysing Fina Miralles' notebooks of the time and a kind of "deep hanging out" with Fina Miralles (Vara 2021b).

During my research I have located a somatic point of view from which I could unpack the body actions by the artists and distinct phenomenological, affective and conceptual perspectives of the gestures. This somatic empathy was useful for approaching my research questions: "What was emancipatory or liberatory about her bodywork process? What might feminist political action look like when it is not overtly revolutionary?" I filmed interviews and situated myself beside Fina Miralles or what Katve-Kaisa Kontturi terms "ways of following" (Kontturi 2018). I listened/felt/sensed the traces of her work, and the artist talking/gesturing about it. To develop this strategy, for instance, I sometimes used the first person to describe what was happening in the performances as a way of putting myself into the action, and to understand what the process of the action was. I also built a vocabulary and lexicon that came from observing and feeling/sensing—in a haptic way—the traces of the performances. I was not so much imposing codes as listening to the data and to my own body experience. In another occasions, I re-performed some actions (Vara 2021b) experiencing in my own body the qualities of the elements (earth, water, contact with a tree, etc.) that the artist used in her performances. This helped me to define the weight of materials and outline, contour or represent the borders of a given body. To situate and experience the movements of my body with similar gestures to those Fina Miralles made, was a method of assembling sensorial information about her corporeal work that would not have been accessible through using only observation, words, descriptions or visual analysis. In re-acting them I expanded and increased my understanding about the feminist emancipatory potential of the corporeal scores, body mapping and kinesthetic knowledge generated through artwork by Fina Miralles. This methodology allowed me to develop a process in the analysis that evolve into a somatic one through kinesthetic empathy.

## 2. Results

### 2.1. On the Sensorial and Corporal Scores

"La pràctica artística ha de ser l'explicació més clara de la ideologia I de l'acte de creació. L'obra d'art ha de ser un process obert per a tothom. Ningú pot gaudir-ne o comprar-ne parcialment. Ha de seguir el procediment més adequate perquè la ideologia sigui exposada el més clara possible"

The artistic practice must be the clearest explanation of the ideology and the act of creation. The art piece must be an *open process for everybody.* Nobody can enjoy or purchase it partially. It must follow an *accurate method* to expose ideology the most clearly possible.

(Fina Miralles 1972–1976, p. 22)

[my translation, emphasis added]

On Fina Miralles' webpage, under the tab "Pertinença" (Belonging), one can find a series of paintings called *Sensitiveland* (1972) (Figure 3, which prefigure the work of *Translacions* and *Relacions*). In these paintings, even just in their titles, the senses are present. These paintings may in fact be seen as "pre-exercises," explorations of the sensorial, which might be instructive for an explicit presence of the body of the artist in the works and a more clearly expressed feminist praxis by the artist. There is an initial exploration of the senses, such as tactile sensation, and the names or colours of sounds. Through colours and forms, Fina Miralles stimulates physical sensations and an aural sensation through letters and words that are onomatopoeic (i.e., words such as "bulzz"). I can hear the sounds and also have tactile sensations when I observe these images. Also it must be noticed that during my conversations with Fina Miralles I realized how much she communicates through music and sounds. Sometimes a sentence can be full of sounds expressing what she is communicating without words. *Sensitiveland* is a group of fifteen paintings and was the first individual exhibition by Fina Miralles in October 1972. This series is closer to a form of haptic visuality for the viewer, and opens a kinesthetic experience, giving us information about her explorations before actually using her corporeal dynamics. According to art historian

Marta Pol, with these paintings "she abandoned the learned culture, the traditional artistic conception, to express the world linked to the senses and feelings" (Pol 2012, p. 40). These works represented a whole world of sensations that could be understood and perceived by the senses.

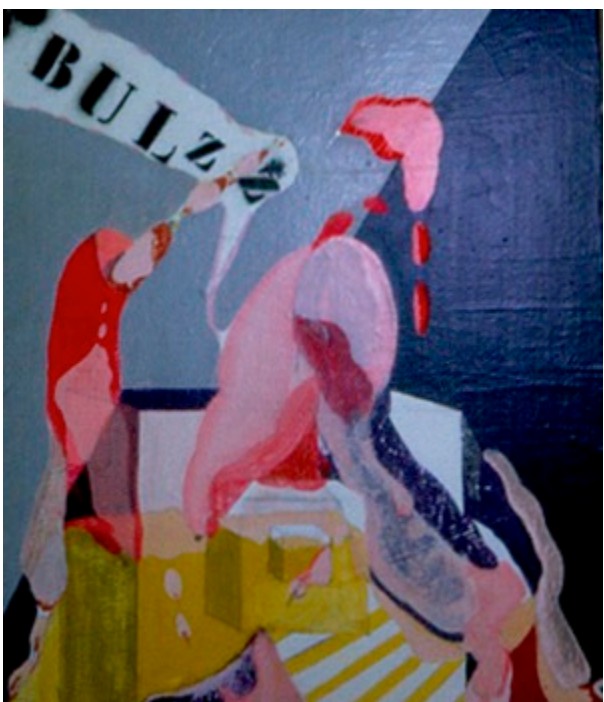

**Figure 3.** *Sensiteland Bulzz* (1972), mixed technique by Fina Miralles. Reproduced with permission from the artist and *Museu d'Art de Sabadell* (MAS) archive.

I appreciate how much of these paintings express the relation between rounded forms (the body and nature) and squared forms (the objects). When I look at these paintings, I can feel the corners and the soft objects and its relations. Fina Miralles paints what she will further develop with her own body and matter in the world in her later series. For this reason, I consider these paintings some of the initial events in the process of becoming physically in a relationship with natural elements and objects around her and leaving tracks. These paintings are the beginning of a search for and within the body that will be manifested more clearly in the next analysed pieces. In the following, I elaborate and read some of her artwork as a search that is related to corporeal perception and kinesthesia.

In the same year that Fina Miralles painted *Sensiteland* (Figure 3), she began to experiment with elements from nature. These will be the kind of materials that she will use to perform the movement of them to spaces to which they do not originally pertain (*Translacions*), or she will actually put her own body *in relation to* them in the series *Relacions*. She developed the installation named *Natura Morta* (Still Life) (1972) within the project *Naturaleses Naturals* (Natural Natures).[7] Fina Miralles displays different elements from nature on a table. Instead of painting them in this case, she presents the elements themselves. In this piece, she not only breaks with art history's traditional parameters of the "still life," but also initiates a sensorial experimentation in her art practice and intervention in the world.

To complement the piece *Natura Morta* (1973) (Figure 4), Fina Miralles gave the public a text in which she explained that, apart from the materials shown on the table (shells, water, algae, sand, wood, grass, stones and leaves), there were other materials present such as the air, a material that the viewer could experiment with outside of the gallery. She asked the public to look at the sky and suggest a series of instructions to appreciate the landscape. In her notebook, Fina Miralles writes: "look at the sky three times a day:

morning, afternoon and night; and four times a year, according to the seasons: spring, summer, autumn and winter."(Miralles 1972–1976, p. 218).

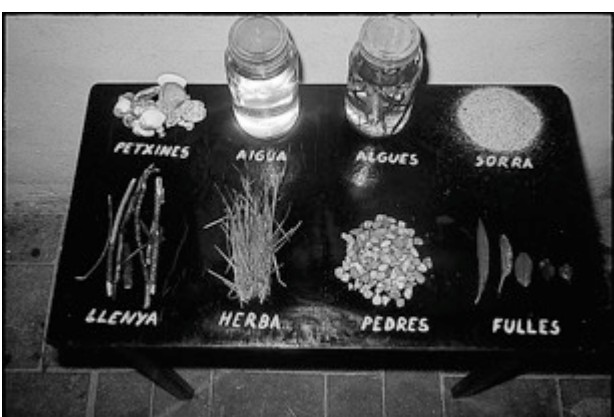

**Figure 4.** *Natura Morta* (Still life) (1973), installation by Fina Miralles. Reproduced with permission from the artist and *Museu d'Art de Sabadell* (MAS) archive.

In this way, the public assumed a role similar to that of the artist because they added their own experience to the work, and were expected to read this installation as a kind of hieroglyph or survival guide. With these proposals, Fina Miralles expanded her field of action outside the physical frame of the room, bringing the viewer into the experience of their daily lives. The work exists less as a stable object than as a generator of perceptive answers. Now the spectators cannot remain indifferent, they are forced to carry out successive acts of perceptive response to the stimulus, even out of the context of the exhibition.

Alexandre Cirici argues that the artist's strategy of giving the viewer an instruction manual is an approximation of the methodology of music:

> With this, she approaches musical composers who do not give us sound images but write a *score* and indicate some instruments. Music is a sensitive reality that may be performed or not by the reader of the score but the author does not play it (but the musicians) (Cirici 1980, p. 43) (emphasis added) [my translation].

Although all the musical content is recorded in the score, in order to produce sound, it is necessary to know how to read the symbols that indicate time, notes and instruments. Cirici, drawing from music, indicates that the text by Fina Miralles is a score, the unfinished symbolic elements that stimulate the productive process of reception-creation. Thus, there is a corporeal score that is instructive and connected with the rest of her work.

In the same project (*Naturaleses Naturals*) there is a film very much connected with the previous piece of *Natura Morta* and the instructions given in a paper to the public. It is really a way to show how to perform those instructions. Fina Miralles shot a 25 min Super-8 film entitled *Fenòmens Atmosfèrics* (Atmospherical Phenomena) (1973) (Figures 5 and 6). This piece, like *Natura Morta*, was within the project *Naturaleses Naturals*. In this film, Miralles shot different elements and natural events. The film records sea waves, the sunrise and sunset, the passage of clouds, the disappearance of a cloud by the action of the sun, the expansive waves created by throwing a stone in the water, the rain on the sea, the wind moving the leaves of the branches of the trees, and the flight of birds.

There is a clear sense of physical touching between the elements, and the sense of being touched between them and by them while embodying this spectatorship. Fina Miralles shows the relations between different elements in nature and this haptic relation between them. The water touches the rocks; the wind touches leaves, etc. The format of the Super 8 film gives a form of thickness to the elements. The film evokes close engagement with surface detail and the texture of the filmed elements. Since this is a piece without sound, the viewer is invited to observe and contemplate the image itself rather than being in a narrative flow. It encourages a corporeal engagement. There is roughness and a cut

between scenes, and this makes the viewer sense a third dimension in the screen, which invites a feeling of being in the landscapes. Observing this film, I felt surrounded by the water, the sunrise, listening to the stone thrown into the water, creating rounded waves. Her film thus constitutes another language for proposing modes of being *in relation with*, as proposed in *Translacions and Relacions*. Fina Miralles does not touch the elements with her body (something that she would do later in her work, as I explain further in the article); however, a connection is established through the very fact of her filming them. *Fenòmens Atmosfèrics* becomes a way of encountering the world through the camera.

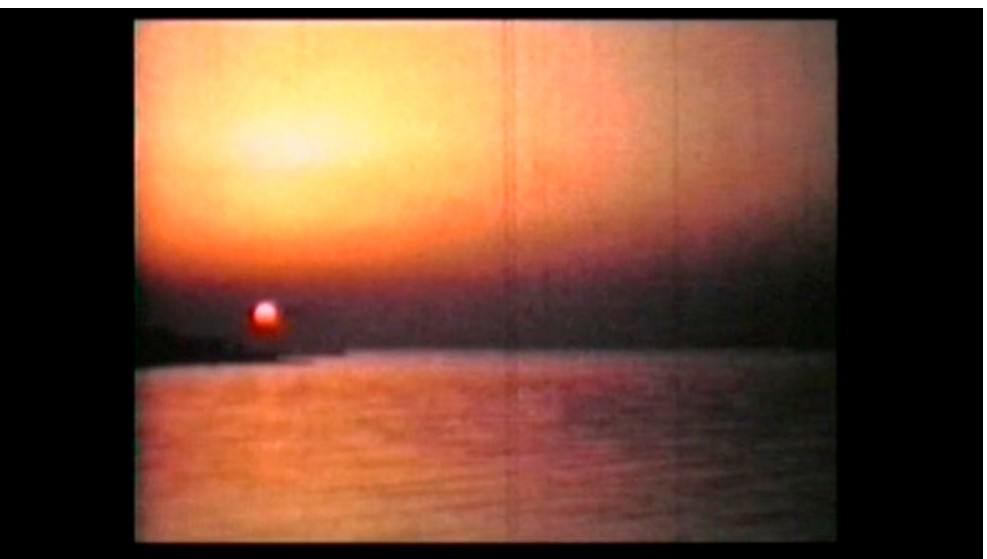

**Figure 5.** *Fenòmens Atmosfèrics* (Atmospherical Phenomena) (1973) still Super 8 film by Fina Miralles. Reproduced with permission from the artist and *Museu d'Art de Sabadell* (MAS) archive.

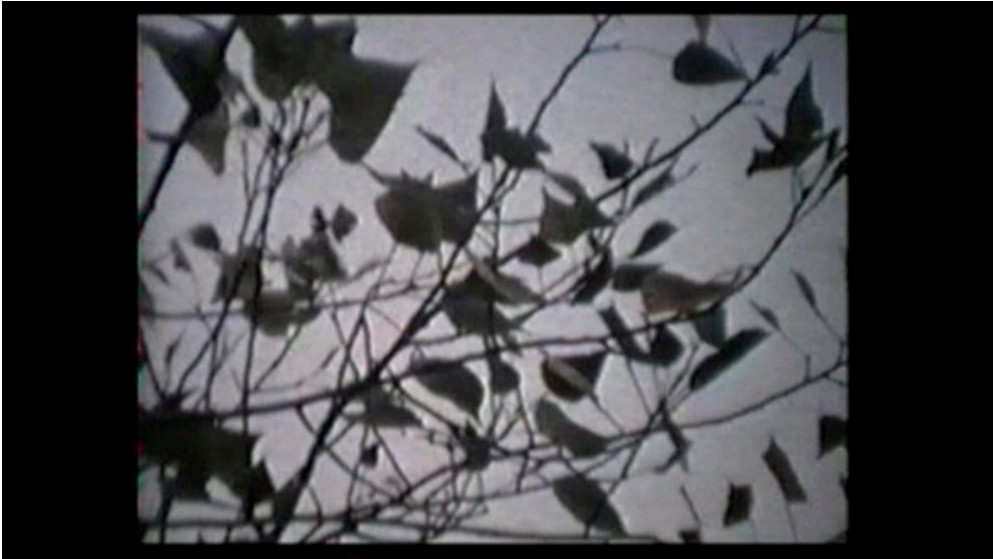

**Figure 6.** *Fenòmens Atmosfèrics* (Atmospherical Phenomena) (1973) still Super 8 film by Fina Miralles. Reproduced with permission from the artist and *Museu d'Art de Sabadell* (MAS) archive.

Prior to filming in her notebook, she writes the word "Films", which corresponds to most of the scenes in *Fenòmens Atmosfèrics*: "the waves when they get to the seashore, how the sun rises, from inside a balcony to shoot how the day light disappears and it gets the night, how the sun dissolves a cloud, how the clouds move because of the wind, how the wind moves the trees, the rain water on the sea, the flight of Plaça Catalunya, how the seagulls fly over the sea, how a bean grows, to shoot a cypress tree with sun and cloud"

(Fina Miralles 1972–1976, p. 11) (my translation). These notes address the importance of *movement* and touch, something very related to the sense of kinesthesia. There is a metonymic aspect in the sense that one thing is in relation with the other, and one thing touches the other. I argue that emotions and stimuli come from the action of looking at the natural elements and the relations between these elements, not only from sight, but from imagining hearing the sound of the waves, the wind, etc. The work of art and the subtraction of natural or habitual perception amplify the possible work of the imagination, expanding our potential.

There is a thread that connects the pieces read in this section (*Sensitive Land*, *Natura Morta* and *Fenòmens Atmosfèrics*). Fina Miralles proposes sensorial experimentations, practices, alteration of usual meanings, tweaking, variations and corporeal exercises to create new ways of perceiving the body and surroundings. Noland states that "kinesthetic experience–the sensory awareness of one's own movement–can indeed encourage experiment, modification, and, at times, rejection of the routine" (Noland 2009, p. 3). Her thinking supports my argument in seeing these earlier sensorial exercises in Fina Miralles' oeuvre as pre-exercises of later work and a presentation of a somatic and perceptual manner to access knowledge and feminist emancipation.

### 2.2. On Leaving Tracks and Mapping the Body

"Tot el que sé es a través del que he vist, del que veig. Jo tinc un coneixement, diguem, no llegit. . . sinó mirat I viscut"

All that I know is from all that I saw, all that I see. I have a knowledge, if you want, 'not read' . . . yet a sight and lived knowledge

(Miralles, personal conversation, 2016–2019) [my translation].

In part of the archive, there are a few photographs (Figure 7) where Fina Miralles is playing with sand and natural elements on the beach. These images do not pertain to official documentation of some of her performances. This is likely one of the moments where she was gathering the materials for the installations (for instance, *Natura Morta*, 1973; or while she was filming the sunset for *Fenòmens Atmosfèrics*, 1973). She gathers knowledge stemming from bodily experience, in this case by making a round and ephemeral track on the sand.

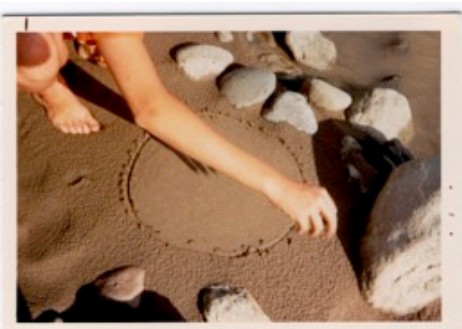 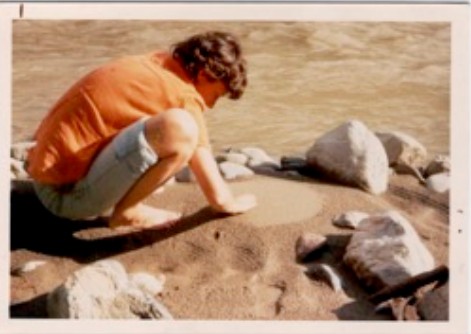

**Figure 7.** Fina Miralles working/playing with materials from nature (1970s). Author: Unknown. Reproduced with permission from the artist and *Museu d'Art de Sabadell* (MAS) archive.

Fina Miralles' explorations imply that the body is gradually appearing through her artistic actions and explorations in the world, which may be read as a search for and through the body. This is related to a construction of subjectivity that starts by certifying the existence of a body to later finding it as a source of knowledge and emancipation. Her whole process in the 1970s offers a model for feminist methodologies for cultivating agency by creating spaces of bodily appearance and somatic sensations that lead to a reconstitution of the body. In the following, I approach the different manners in which Fina Miralles leaves tracks with her body in her work: first, moving natural elements with her body, to

later being covered by soil or leaving the marks of her body in the sand, with her hands, or in the ground. Some of these actions lead to mapping the body.[8] In this way, I argue that Fina Miralles was experimenting with stimulus and feelings and increasing the value of the lived body, certifying its existence and consequently reconstituting it and bringing it to life. That is, an introspective set of corporeal dynamics that confirm the presence of a body—its corporeality—being alive, a body that occupies a space and a position.

In January 1975, Fina Miralles performed the action of covering herself with sand from the beach in Premia de Mar, a small town in Barcelona: *El Cos Cobert amb Sorra* (Body Covered With Sand) (Figure 8). This action takes place on a beach where the sand is fine. The idea of the action was "to cover herself with sand, to leave the body marked on the sand" (Miralles 1972–1976, p. 24). Sand, like stone and earth, is a solid material, but it differs from other previous materials utilized such as the stones because it is small and compact. It appears that Fina Miralles wanted to experience a different weight on her body, a subtler one. Also, her notebook indicates an intention to leave a mark, a bodily trace, in these works. The fact that the sand is thinner and has less weight than the other materials allows Fina Miralles to get rid of the material in a single movement. The process of covering occurs from feet to head. The body is covered with sand, over her clothes, to her head.

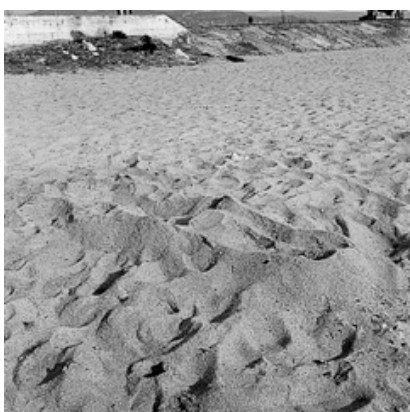

**Figure 8.** Black and white photographic documentation with the trace of Fina Miralles' body after the action *El Cos Cobert de Sorra* (Body covered with sand) (January 1975) by Fina Miralles. Reproduced with permission from the artist and *Museu d'Art de Sabadell* (MAS) archive.

It is important to emphasize that when she experiments with a different material, she obtains a result that is different not only visually but also perceptually in her body. For instance, in *El Cos Cobert amb Sorra* (Figure 8) the artist could stand up in only one or two movements; therefore, the material that she had on her body was deposited to the sides more rapidly. Thus, she could highlight her silhouette, that is, the hole produced with the process of being covered and standing up. The artist's body mark remains—ephemerally— in the sand.[9] The final photograph is the shape that Fina Miralles leaves in the sand after she stands up from being covered (Figure 8). This silhouette could be confused with the small dunes that are carved in the sand by the wind.

While in *El Cos Cobert amb Terra* (Figure 9) the last document is the shape of Fina Miralles' body under the soil, in *El Cos Cobert amb Sorra* I find another interest in the documentation: leaving the trace of her body in the sand. It is an opposite process of shaping the body: one is the lump of her body under the soil and the other the trace of her body in the sand. I suggest that the role of this documentation is to map the body, exploring the borders and the spaces that her body can occupy, and this also becomes a transformative experience that builds on corporeal agency. The very fact of marking and leaving tracks facilitates some occasions of a subversive practice in that given socio-political context. These are also part of the experimental exercises through which Fina Miralles explores the sensorial as access to the world through the borders of the body, through leaving those borders marked, for instance, in the sand.

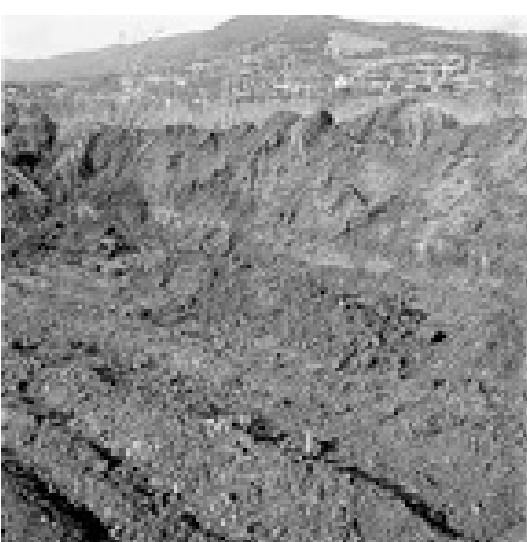

**Figure 9.** Last black and white photographic documentation of the action with the shape of Fina Miralles' body under the soil in *El Cos Cobert amb Terra* (Body covered by soil) (January 1975) by Fina Miralles. Reproduced with permission from the artist and *Museu d'Art de Sabadell* (MAS) archive.

In the series *Translacions* there is also an intention of corporeal exploration by leaving marks with her body or with elements. In *Duna* (Dune) she moves the sand of a dune to an orchard. She does tactile work. She uses the shovel to move the sand to the orchard and builds a path of sand, a mark in the orchard. At the end of this path she builds a little mountain of sand. This action was filmed in Super 8 and there are a few close-up shots of Fina Miralles' hands where she takes sand, builds the little mountain, and pats it to make it uniform.

In *Duna*, Fina Miralles makes the shape with the sand (she leaves the marks of her fingers and hand on the sand: Figure 10) and in *Relacions* she is under this sand in order to make the shape. *Duna* is a preparatory action of creating traces with materials that helps to develop the subsequent process of leaving her own body trace in *Relacions*. I could say, using Noland's argument, that she is carrying out an "exploratory project" in investigating her bodily image. As Noland affirms in "*Agency and Embodiment*:"

> To be sure, the gestural routines of inscription are violently disciplining; they can shape and suppress the individual body by submitting it to highly rigorous standards of execution. However, *the process of making marks also offers opportunities for subversion*: we can leave our marks in the wrong place, invent private or countercultural mark systems, or use mark-making as an *exploratory project*, investigating how our bodies might move differently and thereby achieve materialization and cultural legibility in unexpected ways (Noland 2009, pp. 214–15) (emphasis added).

This process of making marks in unexpected situations continues with the action *Relació del Cos Humá amb l'Herba i la Impronta del Cos Sobre l'Herba* (The Body in Relation To the Grass and the Trace of the Body in the Grass) (1974). In this action, Fina Miralles leaves the mark of her body on top of a surface. In her notebook she writes: "Cover with straw and leave the mark of the body" (Miralles 1972–1976, p. 24). In this case, the body's shape leaves a mark on the grass as a result of crushing it. The idea of working with dry grass that covers her in *El Cos Cobert de Palla* (Body Covered with Straw) (addressed above), generates a different form than that of the body imprinting her shape on the grass, since the living grass, as a material, has a plasticity that is completely different from dry grass. There were three images of this action: one of the field (left-hand side, Figure 11), the next one with Fina Miralles lying on the field (centre, Figure 11) and the last one with the traces of her body (right side, Figure 11).

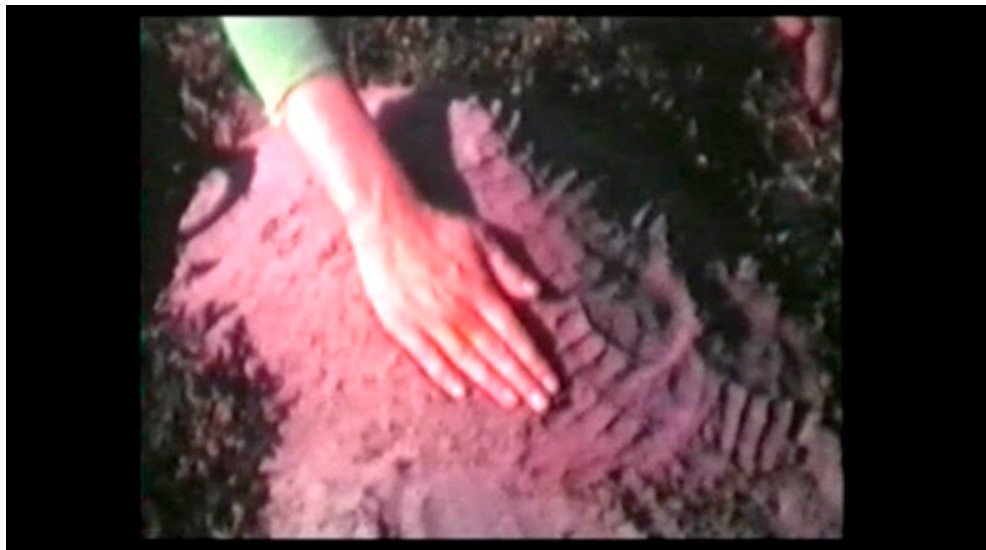

**Figure 10.** Super 8 film documentation (Author: Unknown) of *Duna* (Dune) (18 November 1973). Still Super 8, film by Fina Miralles. Reproduced with permission from the artist and *Museu d'Art de Sabadell* (MAS) archive.

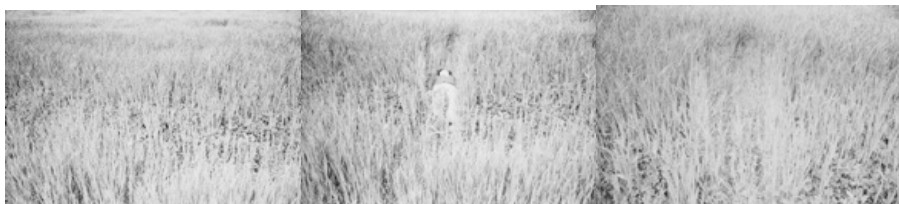

**Figure 11.** Black and white photographic documentation of *Relació del cos humá amb l'herba i la impronta del cos sobre l'herba* (The body in relation to the grass and the trace of the body in the grass) (April 1974) by Fina Miralles. Reproduced with permission from the artist and *Museu d'Art de Sabadell* (MAS) archive.

Fina Miralles describes this material in her notebook as follows: "The grass is solid, nothing compact and extremely flexible" (Miralles 1972–1976, p. 21). We observe again this detailed research into the characteristics of the elements she is using to put in relation to her body. This particularity of the material in its natural context, for instance the grass in a field, allowed the artist to work plastically once again on the idea of leaving the negative footprint of her body on top of the material. While making traces, Fina Miralles is experimenting with the sense of kinesthesia and mapping the body. These pieces are the beginning of an exploratory method and its value to feminist politics. That is, the fact of mapping a body through sensorial explorations and the particularity of her body becoming a site of displacement. This requires a body that is involved in processes of transformation and change, evolving into an active agent. The body, thus, plays a fundamental role in feminist political thought with its own weight and volume, in movement and occupying a position.

Two and three years after of these series of work, Fina Miralles developed *Petjades* (Figures 12 and 13). This piece was made one year after the death of the leader of the Spanish dictatorship. In this piece, Fina Miralles walks slowly, with two shoes printing her name on the ground. There is a connection with the previous sensorial explorations. The fact that she wanted to print her name makes her develop another type of movement and corporeal consciousness. This movement becomes an exercise/practice in a specific awareness of her body walking in the street: a body that comes to be in relation with itself and the world. However, *Petjades* is different from the previous work I discussed in this article. This performance is operating discursively and symbolically.

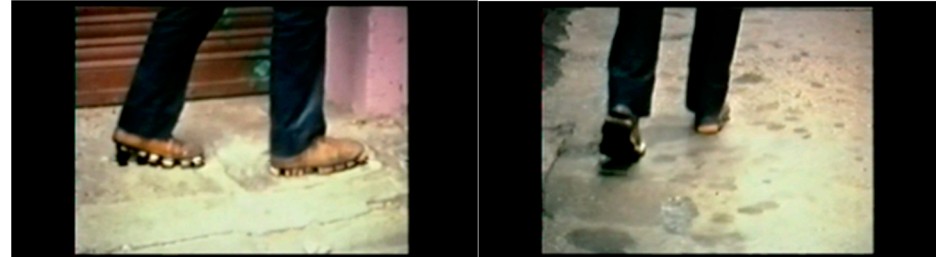

**Figure 12.** Two video stills. Super 8 film documentation (Author: Unknown) of *Petjades* (Footprints) (1976) by Fina Miralles. Reproduced with permission from the artist and *Museu d'Art de Sabadell* (MAS) archive.

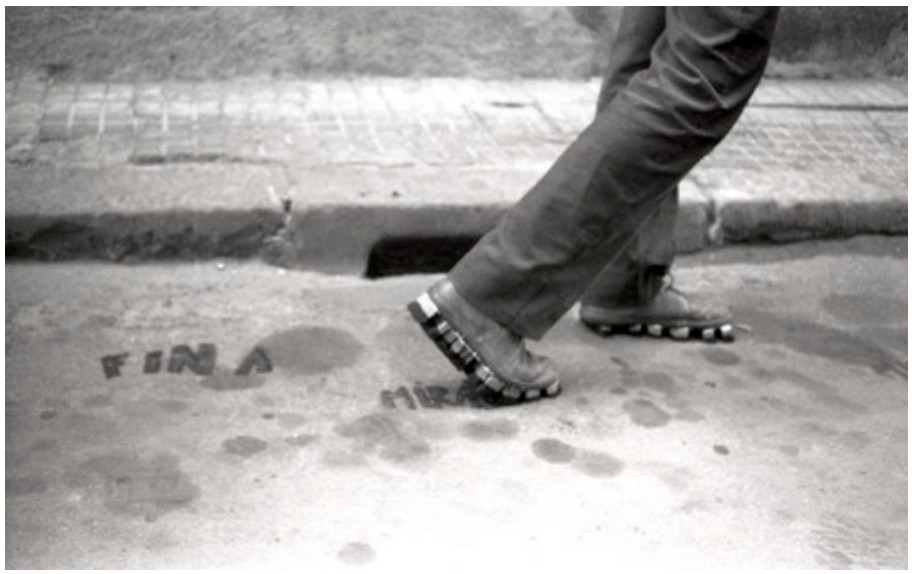

**Figure 13.** Fina Miralles inscribed her name on the ground in the streets of Barcelona. Documentation in black and white photograph of *Petjades* (Footprints) (1976) by Fina Miralles (1976). Photograph by Carles Raurich. Reproduced with permission from the artist and *Museu d'Art de Sabadell* (MAS) archive.

This performance reflects on public and private space and its different occupation regarding gender structures. She walked through the streets of Barcelona, leaving marks on the ground with her shoes—her name and surname. This was a pioneering political work for the personal occupation of public space by a woman stamping her own name (and her single name at that), and at the same time an incorporation of her previous corporeal dynamics. If in the previous actions analysed in this article I observed a proto-political work—-parallel to the feminist activism at that moment in Spain–in *Petjades* I see a continuation of that procedure and a connection with these previous works. Fina Miralles joined an interoceptive process of recuperating the body with a more overtly political work denouncing the situation of women in the Spain at that moment. She traced a path with her printed name while parodying the power system of capitalism and patriarchy. In the film, she used her voice-over for the first time in a performance, and said:

> In the city, we clearly face the characteristics of our capitalist society. The power in the sense of ownership is deeply rooted in our way of life, our behaviour, our organization and our laws. Not only do we legalize our personal assets to acquire power (house, car, land, objects. . .) but also the people who are under our protection. My wife, my son, giving the name to reaffirm that they are our property and are under our power. From here, the shoes that I present with my name on the sole like an office rubber stamp, and the action of printing my name in all the places where I step, are simply to highlight this possessiveness

and power, overreaching things to the ridiculous, the incredible, saying that the place where I walk is "mine", "it's mine" (Voice-over by Fina Miralles in the film *Petjades*, 1976) (my translation).

*Petjades* confirms but also challenges some of the tendencies that I read in her oeuvre prior to this period. I see in *Petjades* a culmination of a set of practices that I identify as feminist, even if they are not explicitly so. Therefore, I argue for continuity between the previous works explained in this article and *Petjades*, a clearly political work. I also want to specifically note how scholars ignore the kinesthetic and sensorial aspects of *Petjades* because of its political and semiotic valence. In this sense, the concepts of kinesthetic knowledge and corporeal agency mobilized and fleshed out throughout the analysis of *Translacions* and *Relacions* culminate in a more explicitly political and feminist practice in the post-Franco era that also is shaped by kinesthetic aspects. The type of movement that Fina Miralles develops in *Petjades* increases her kinesthetic consciousness. While watching the documented performance, it is possible to feel its haptic qualities and how they might produce a certain effect that is not simply visual but also sensory, not only an observation of Fina Miralles in the process of walking and leaving a mark, but also something that the contemporary spectator may feel. This reading derives in part from the performative aspect of the gesture, which remains even as she imprints her name in the street. Fina Miralles reclaims her body in the public arena through a sensorial relation with her surroundings, in this case the relation that her feet establish with the very texture of the ground she is in relation with. She presents the space occupied by walking and reinforcing the space within the body.

Kinesthetic experiences reinforce this consciousness within the body. She creates a place inside and outside herself, *engaging the self* through the *sensorial*, walking in a public domain, as a woman, as a subject in society. The idea of taking control over one's body, claiming her subjectivity and gaining space for corporeal agency within the body, albeit momentarily or precariously, is something that I have identified in this piece and other performances addressed in this article. Fina Miralles reclaims her body and I would say her own sensorial spectrum in relation with the matter around her and her own material body, a key aspect in the process of building agency. I have drawn from Noland (2009), Sheets-Johnstone (2016) and Sklar (1994, 2008) for the connection they establish between kinesthesia and agency, which was particularly useful for reflecting on the emancipatory possibilities of the corporeal work carried out by Fina Miralles under the dictatorship. These simple bodily actions show the possibility of bodily knowing and a feminist liberatory intervention emerging in the context of the repressive gender order of the dictatorship.

She engages in a particular material search to take over public space claimed by the feminist movement, and also encourages an attention within the body through that very relation with the materials around her. Putting all her senses into this walk is acting as a tool for self-exploration and kinesthetic experience, and, thus, as an activist intervention in a public space. She could be asking herself: what does it mean to put "myself" into what I walk on? How does it feel to engage my *self* through the sense of walking with agency? While she traces her wet footprints with her name on the ground, I believe what she means to say is "I am alive, I know who I am, I am here, I am a subject with agency".[10]

## 3. Conclusions

If feminism has frequently understood the body as a site for liberation from restrictive control but not a *source* for that emancipation—due to the understanding of agency under Cartesian paradigms (Liimakka 2011, 2013)—then what impact do Fina Miralles' actions have for comprehending a corporeal or somatic agency that contributes to building methodologies for emancipation? Fina Miralles presents her work as explorations of the self through movement and kinesthesia, subtle transgressions under the Franco dictatorship and emancipatory feminist somatic dynamics.

Fina Miralles' methods are exercises and practices defined by corporeal strategies. There is an emphasis on process, and also on the fact that such a process must be available

to everybody. Her actions are *body compositions* lopen to being interpreted. In this sense, I argue that Fina Miralles is offering us *emancipatory corporeal scores*, and this can be read as a feminist intervention. I see these scores of freedom as instructions for noticing the body through kinesthesia. They are scores for becoming aware of the subtle movements in our body, even if it is a still body, such as those that augment the perception of tendons, ligaments or muscles. What is freeing here is the awareness of the body through subtle movement within the body. I consider this feminist because of the particular feminist politics from which I draw in this article, which consider the body as the site of inscription and the main source of knowledge and emancipation. Through the corporeal instructions/scores that Fina Miralles offers, it is possible to notice the sensorial in a subtle way and this leads to a potential reconstitution of the body.

This has an impact on feminist liberatory practice because of three main aspects: (a) the body becomes a space of corporeal inscription and appearance, (b) there is a somatic awareness, and (c) this corporeal consciousness leads to a reconstitution of the body under dictatorship. I read this as specifically feminist because of the connection I make between kinesthesia as a manner of embodiment and agency, and a feminism that centres its attention on corporeal processes. Fina Miralles' emphasis on mapping the body, tracing its tracks, and the gradual appearance and emergence of it accentuates a manner in which corporeality can be perceived through somatic exercises and practices, and this presents the body as a site for agency and liberation. If the corporeal is the starting point for processes of agency, then Fina Miralles' practices and exercises are a feminist guide for emancipatory methods that begin in the sensorial body. If feminism has focused its attention on the body as a site of oppressive power, then Fina Miralles' actions offer an avenue to be explored regarding the body as a source of liberation for women.

**Funding:** This research was funded by Fonds de Recherche du Québec—Société et culture (FRQSC).

**Institutional Review Board Statement:** The members of the University Human Research Ethics Committee (Concordia University, Montreal, Quebec, Canada) have examined the application for a grant to support the project "Emancipatory Actions: Video and performance by Fina Miralles (b. 1950, Sabadell, Catalunya)" (Certification Number 30005621—2016–2019), and consider the experimental procedures, as outlined by the applicant, to be acceptable on ethical grounds for research involving human subjects.

**Informed Consent Statement:** Informed consent was obtained from all subjects involved in the study.

**Conflicts of Interest:** The author declares no conflict of interest.

## Notes

[1]   Franco was a Spanish general who ruled over Spain as a military dictator from 1939, after the Nationalist victory in the Spanish Civil War (1936–1939), until his death in 1975. This period is called Francoism. The establishment of the Franco regime implemented the so-called National Catholic system, based on an ideology that combines elements of Catholic conservatism with elements of the new political current that emerged in interwar Europe (fascism). National Catholicism (*nacionalcatolicismo*) was a sign of the ideological identity of Francoism. Its most visible manifestation is the hegemony that the Catholic Church had in all aspects of public and even private life. Gender relations were defined in terms of the values of national Catholicism. The female ideal of *nacionalcatolicismo* was that of the submissive woman, daughter, wife or mother, condemned to domesticity, relegated to the private sphere of the home and the family, and to caretaking. In this new configuration of feminine identity, motherhood was glorified. This was a key piece of repression. Women were barred from access to the public sphere and, therefore, to the space of politics, which was considered a male space. If women were politically active, they were considered to be challenging their gender roles and their ascription to private space that was considered natural for Catholicism (Raquel Osborne 2012).

[2]   Second-wave feminism emphasized shared feminist consciousness and feminist resistance to the objectification of women. Consciousness-raising groups were frequently based on a traditional definition of agency as resistance (and thus a Cartesian understanding of agency). A researcher in Social Psychology, Satu Liimakka, has explored women's bodily experiences and possibilities for embodied agency. She suggests that a strategy of the mind, such as being conscious of the oppressive status for women, is not enough to provoke a change in learned corporeal roles. She proposes an agency that arises from and in the body (Liimakka 2011, 2013).

[3]   During the late years of the Franco regime (and in the transition to democracy), art became political work through the performativity of the body in forms and spaces where it was hitherto unknown or unprecedented. These actions—even though sometimes

not analysed from a feminist perspective—have an importance for feminist politics in a moment where the control of bodies was an essential tool of the dictatorship. There are several works of performance art that propose mapping the body through different strategies in relation with materials or positions of the body and that I read as a feminist intervention: Esther Ferrer (b. 1937), Àngels Ribé (b. 1943) and Olga Pijoán (b. 1952–1997). However, it is important to note that the feminist movement which emerged in the context of Francoism was something parallel to and separate from artistic manifestations (Bassas 2013).

4   *Translacions* (Movements) (Vara 2021a) is a Catalonian word that means to move something or someone from one place to another. It is also the elliptical motion that describes the Earth's movement around the Sun. *Translacions* is a series of actions in which Fina Miralles moves elements to an environment where they initially do not belong: *Deixada Anar de Cargols* (Letting Snails Free) (1973), where she paints snails in groups and liberates them in a square in Parc de la Ciutadella, Barcelona, Spain on 1 November 1973; *Flotació d'Herba en el Mar* (Fleet of Grass in the Sea), performed in Premià de Mar on 11 November 1973; moving sand to a field in *Duna* (Dune) in which she moved sand from the beach of Sant Marti d'Empúries to a farmland on 18 November 1973; and *Dona-Arbre* (Woman-Tree), in which the artist's own body is transplanted to the middle of the landscape of Sant Llorenç del Munt, Barcelona (Vara 2019).

5   Between March 1974 and January 1975, Fina Miralles performed a series of actions called *Relacions* (Relations). This series was divided into two subseries: *Relacions del Cos, Elements Naturals* (Body Relations, Natural Elements), where the artist covers and uncovers herself with different natural elements and *Relacions. Accions Quotidians* (Relations. Everyday actions) where the artist performs quotidian activities.

6   Fina Miralles developed a group of actions that have attracted more attention from researchers (Assumpta Bassas, Maia Creus, Patricia Mayayo, Juan Albarrán Diego and Juan Vicente Aliaga) who have observed a feminist component in her work. In the Spanish literature on feminism and art, there are references to pieces by Fina Miralles, particularly those works that had a clear activist purpose against abuses of power, which also deconstructed the female role under Francoism: *Standard* (1976), *Enmascarats* (Masked) (1976), *Triangle, Simbología de Poder i Mort* (Triangle, Symbols of Power and Death) (1976) and *Matanzas* (Slaughter) (1976).

7   The *Naturaleses Naturals* exhibition was first presented in 1973 in Sala Vinçon in Barcelona and then in 2016 at the *Museo Arqueológico Natural*. In this project, there is an emphasis on the relation between natural and artificial materials. Fina Miralles explains (personal conversation, 2016–2019) that in this exhibition, she was showing natural materials without transformation (challenging the ego figure of the "artist" that "transforms" matter), artificial materials (imitation grass, etc.) and live animals with stuffed animals (here she carried out an action with dove—alive and stuffed—in which she opened a cage and only the ones that were alive started to move and leave). For this last action, Fina Miralles wanted to highlight the difference between being alive and being dead. Also, she mentions a poem from a Basque poet, which she put near the cage: "If you would have cut the wings, it could be mine, yet it would have stopped being a bird." This interest in showing "movement" and "liveness" is also an exploration of the dictatorial repression of the bodies and her interest in developing liberatory dynamics through movement.

8   Body mapping has been proved to be a therapeutic tool that mixes bodily experience and visual artistic expression (Immadisetty 2012) used in research, therapeutic, and educational contexts (De Jager et al. 2016). There is research carried out regarding "whole-body mapping," defined as the tracing of a person's body to create a "life-sized outline, which is filled in during a creative and reflective process, producing an image representing multiple aspects of . . . their embodied experience" (ibid. p. 1). According to these authors, there is a necessity to further research how "body mapping" is a useful research method to produce and disseminate knowledge, and more investigation would be necessary to determine the value of "body mapping" to research and the clinical, educative and political spheres.

9   These strategies of covering–uncovering and leaving tracks remind me of the Cuban–US artist Ana Mendieta's performances *Feathers on a Woman* (1972), *Blood and Feathers* (1974), *Tree of Life* (1977), *Burial Pyramide, Yagul, Mexico* (1974), and *Silueta* (1973–1977). Mendieta also relies on and encourages a kinaesthetic experience.

10   These sentences, written by myself, are part of my methodology and inspired by the performance by Fina Miralles, *Recordant aquell temp tan gris* (Remembering those grey times) (2015), where she covers her body like a mummy and then uncovers it in order to remember the Franco dictatorship, while reciting: "I do not know who I am, I am nobody, I do not think anything".

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
