# Peer review of "Kinesthetic Experience: Emancipatory Corporeal Scores"

_humanities, doi:10.3390/h12060138_

Round 1
Reviewer 1 Report
Comments and Suggestions for Authors
Thank you very much for the opportunity of reviewing this interesting article.
The article offers an original analysis of the corporeal practices by the Catalonian artist Fina Miralles. The artist Fina Miralles proposes sensorial experimentations and corporeal exercises to create new ways of perceiving the body and surroundings.
The article is an attempt to reconsider the value of Fina Miralles’ artistic explorations through the body as a form of kinesthetic processes that reinforce the consciousness within the body, related to the construction of subjectivity and a unique source of knowledge and emancipation.
I think the manuscript is well structured and well written. The abstract clearly reflects the objectives of the studies and main findings. The Introduction also is a very interesting.
The article is precisely framed with clearly defined concepts and categories. The results and conclusion are clearly presented.
I believe the article will be of interest to a wide audience and I consider it to be acceptable in its current form.
Author Response
Thank you very much for taking the time to review this manuscript. I very much appreciate your comments and appreciation of the original analysis of the corporeal practices of Catalonian artist Fina Miralles.
I have answered on the space given.
Thank you in your interest in considering this article acceptable for publication.
Reviewer 2 Report
Comments and Suggestions for Authors
I enjoy reading this article, which is a critical elaboration on a creative and political practice. The reading of Fina Miralles' work in the 1970s is sensitively made, and clearly presented. The only thing I would like to suggest is a bit more clarification - the introductory sections mention embodied research methods, and it is not clear in the article how the embodied research methods have contributed to the reading that is being presented in the article. This clarification will help readers to understand better the way the final reading has come about.
The research tries to propose a reading of Fina Miralles' works in the 1970s through a kinesthetic framework, in order to illustrate how her works are opening the study of female bodily presence as a construction of autonomy. This is highly relevant in the context of feminist agenda, not only for the 1970s, but also today. The topic is original in the sense that the researcher-writer is re-creating the artistic experience as a tool for the critical reading. This personal, or embodied approach is innovative and individual. The paper is well written, with clear organisation and coherent in the presentation of the interpretation and the reading of the artworks. The conclusion is consistent with the evidence and arguments presented. In fact, the main body of the paper already presents the arguments well and carries parts of the conclusion.
Author Response
Thank you very much for taking the time to review this manuscript. I very much appreciate your interest in the analysis of this corporeal creative and political practice. Thank you for highlighting that this research is clearly presented.
Thank you for appreciating my work in this research regarding the "female body presence as a construction of agency". It is my purpose to address from a feminist perspective how corporeal agency it is a key of emancipatory processes.
Thank you for your comments regarding the methodology. I am adding an explanation in the manuscript that builds on the understanding of my methodology. I am adding writing that will explain how the embodied research methods originally contributed to understanding in an embodied way the somatic liberatory processes. This is at the end of the introduction in the new manuscript.
I have added a few paragraphs in the introduction with further information about the methodology (this is also and “academic editors” suggestion: “1 Clarification in the introduction”). I have give details about some of the embodied research methods I used. I do that in the final paragraphs of the introduction. There is also reference to a previous article (2021b) I wrote that explains a more in-depth approach to my methodology. These references are in this manuscript as “X” for blind peer-reviewed purposes.
Reviewer 3 Report
Comments and Suggestions for Authors
Please see uploaded file.

Author Response
To special issues of Humanities (ISSN 2076-0787)
Subject: changes for the editors’ and referees’ approval Special
Issue “With-In Bodies: Research Assemblages of the Sensory and the Embodied”
October 31st , 2023
Thank you very much for taking the time to review this manuscript. I am writing regarding the suggested changes in my manuscript that you can find in the article:
- Reviewer 1: I have answered on the space given.
- Reviewer 2: I have added a few paragraphs in the introduction with further information about the methodology (this is also and “academic editors” suggestion: “1 Clarification in the introduction”). I have give details about some of the embodied research methods I used. I do that in the final paragraphs of the introduction. There is also reference to a previous article (2021b) I wrote that explains a more in-depth approach to my methodology. These references are in this manuscript as “X” for blind peer-reviewed purposes.
- Reviewer 3: I have added in the introduction more details about what was happening politically during Francoism and how it affected women. Since this is not the purpose of the article, the reviewer is suggesting doing this in 2-3 sentences and adding scholars who talk bout it in depth. I do that through the “footnote 1”.
- Reviewer 3: Regarding this comment: “maybe some examples of how other activist/artists responded to this”: This is responded in “footnote 2” mentioning three artists. However, it is important to note that feminist movement in Spain (unless North America) was parallel and separate from artistic manifestations and artists were not considered feminist themselves.
- Reviewer 3: I have approached smaller notes and minor types. As suggested by the editorial “Regarding the reviewer's suggestions, you can keep them as they are, it will go through our professional editing in case it is accepted for publication. Regarding Miralles', we would suggest to keep that format” (Email: 30 Oct. 2023)
- Reviewer 3: Changes annotated in number of lines. Please see the new manuscript with all these suggested changes.
Please let me know if your have any doubts or further details in this manuscript. Thank you.